# SARS-CoV-2-Neutralizing Antibody Response and Correlation of Two Serological Assays with Microneutralization

**DOI:** 10.3390/vaccines11030590

**Published:** 2023-03-03

**Authors:** Amal Souiri, Sanaâ Lemriss, Bouchra El Maliki, Hamadi Falahi, Elmostafa El Fahime, Saâd El Kabbaj

**Affiliations:** 1Laboratory of Research and Medical Analysis of Gendarmerie Royale, Department of Biosafety PCL3, Rabat 10100, Morocco; 2Faculty of Medecine and Pharmacy, University Hassan II, Casablanca 20250, Morocco; 3Laboratory of Research and Medical Analysis of Gendarmerie Royale, Laboratory of Immuno-Serology, Rabat 10100, Morocco; 4Supporting Unit for Scientific and Technical Research, National Center for Scientific and Technical Research, Rabat 10102, Morocco

**Keywords:** SARS-CoV-2, serology, neutralizing antibody, COVID-19, vaccination

## Abstract

SARS-CoV-2 has caused a huge pandemic affecting millions of people and resulting innumerous deaths. A better understanding of the correlation between binding antibodies and neutralizing antibodies is necessary to address protective immunity post-infection or vaccination. Here, we investigate the humoral immune response and the seroprevalence of neutralizing antibodies following vaccination with adenovirus-based vector in 177 serum samples. A Microneutralization (MN) assay was used as a reference method to assess whether neutralizing antibody titers correlated with a positive signal in two commercially available serological tests:a rapid lateral flow immune-chromatographic assay (LFIA) and an enzyme-linked Fluorescence Assay (ELFA). Neutralizing antibodies were detected in most serum samples (84%). COVID-19 convalescent individuals showed high antibody titers and significant neutralizing activity. Spearman correlation coefficients between the serological and neutralization results ranged from 0.8 to 0.9, suggesting a moderate to strong correlation between commercial immunoassays test results (LFIA and ELFA) and virus neutralization.

## 1. Introduction

Severe acute respiratory syndrome coronavirus 2 (SARS-CoV-2) appeared in late 2019 in China and causes COVID-19 [1]. This is a potentially fatal infection with severe immunopathology in the respiratory system [2]. The virus has since spread across the world inducing more than 6.8 million deaths [3] and creating a significant burden on healthcare infrastructures and global economies. Natural SARS-CoV-2 infection generates an antibody response targeting nucleocapsid (N) and spike (S) proteins, including the receptor-binding domain (RBD) of the S protein. Before the introduction of SARS-CoV-2 vaccines, a serological test could be used to identify past infection by detecting any of the SARS-CoV-2 viral protein antibodies. The majority of the available vaccines introduce genetic information in the form of a nucleic acid-encoding SARS-CoV-2 spike protein into host cells. The generated spike protein can then induce binding antibodies to the spike protein and neutralizing antibodies (NAbs). Vaccinated individuals with no history of infection can only test positive for the vaccine protein targets [4]. Otherwise, not all binding antibodies can neutralize the virus because they recognize antigenic determinants that are not involved in the virus entry. Therefore, the detection of neutralizing antibodies is of major significance, since they block attachment of the S protein RBD to the cell surface receptor angiotensin-converting enzyme 2 (ACE2), preventing viral entry and replication [5].

There is considerable interest in identifying SARS-CoV-2 NAbs for measuring immune status and assessing vaccine responses. The neutralizing assay is regarded as the gold standard method to measure functional NAbs [6], although it is quite cumbersome, time-consuming and has not been standardized. Little is known about the relationship between SARS-CoV-2 immune response and NAb responses. A few studies have reported that anti-SARS-CoV-2 NAb titers could have some relation with anti-RBD IgG and IgM antibody levels. Anti-SARS-CoV-2 IgM antibodies reach a peak within 3 weeks and then start to decrease rapidly, while IgG antibodies remain elevated for a long time. Moreover, the correlation between anti-N IgG antibody levels and NAb titers exhibit inconsistence [7].

Given the important penetration of serological rapid tests for the detection of specific anti-SARS-CoV-2 antibodies, mostly immunochromatographic and other automatized assays in Morocco and that have received marketing authorization by the Ministry of Public Health, it is necessary to study their serologic diagnostic accuracy and their performance in detecting vaccination-associated anti-SARS-CoV-2 Ab seroconversion in healthy and convalescent individuals.

In Morocco, the national immunization program that began in January 2021 gives priority to those on the front lines, such as medical staff, national authorities, security forces and those involved in the national education system, as well as the elderly and people vulnerable to the virus [8]. Health service workers including laboratory staff may come into contact with patients with COVID-19. It is important to note that the seroprevalence of anti-SARS-CoV-2 antibodies in health facilities may give an overview of the effectiveness of prevention and control measures.

This paper aims to study the seroprevalence of neutralizing activity and the concordance between two commercial SARS-CoV-2 antibody detection tests, which are not designed to specifically detect neutralizing antibodies, and the microneutralization assay using 177 sera from healthy and convalescent laboratory employees after vaccination campaign. This comparison was made at qualitative and quantitative levels.

## 2. Materials and Methods

### 2.1. Specimen Collection

The total number of laboratory workers participating in this study was 177, comprising 138 PCR-negative individuals and 39 recovered COVID-19 patients. The convalescent participants were diagnosed as COVID-19 positive during late 2020 and the first quarter of 2021. All PCR results, negative and positive, were recorded by routine testing conducted in the same period for symptomatic individuals and contacts. All participants received two doses of the ChAdOx1 nCoV-19 (AstraZeneca, Oxford) vaccine. Serum samples were collected during April and May 2021, approximately 2 months after the last dose of the vaccination administered during the vaccinationcampaigns launched on 29 January 2021 and 10 February 2021, according to the participants’ ages. All participants were negative in PCR during the serum collection. Laboratory personnel were invited to participate and were informed about the purpose of the study.

The participants gave oral informed consent and were informed that the study results would not influence any clinical decisions about their specific case.

Blood samples were taken by trained healthcare personnel. The sera were processed by centrifuging at 3000× *g* for 10 min at room temperature, and were used to assess antibodies against SARS-CoV-2 usingthree different methods: microneutralization (MN) assay and two commercial assays, namely, COVID-PRESTO^®^ (AAZ-LMB, Boulogne-Billancourt, France) a rapid lateral-flow immunochromatographic assay (LFIA), and VIDAS^®^ SARS-CoV-2 IgG (9COG), an automated enzyme-linked fluorescent assay (ELFA) performed in VIDAS instrument (Biomérieux, Marcy-l’Etoile, France).

### 2.2. Rapid Lateral Flow Immune-Chromatographic Assay (LFIA)

The sera were screened for the presence of anti-SARS-CoV-2 antibodies using a rapid lateral-flow immunochromatographic assay (LFIA), COVID-PRESTO^®^ (AAZ-LMB, France), targeting immunoglobulin-M (IgM) and immunoglobulin-G (IgG) anti-S and anti-N antibodies. The assay results were provided within 10 min and positive results were photographed. Scores from 0 to 4 were attributed to each band of IgG according to line intensity: no visible line (negative), faint line, faint band, weak band and clear band respectively.

The band intensity was read by two independent operators who were trained to score the intensity from the pictures of each value (Figure 1). As previously described, this scoring was performed for research purposes to capture semi-quantitative data about the rapid test readout and the reproducibility of subjective interpretation, considering that these are the major analytical factors that affect test performance [9].

### 2.3. Enzyme-Linked Fluorescence Assay (ELFA)

The VIDAS^®^ SARS-CoV-2 IgG (9COG) (ref.423834) assay is a semi-automated qualitative assay run on the Vidas instrument (bioMérieux, Marcy-l’Étoile, France), using the ELFA (enzyme-linked fluorescent assay) principle to detect IgG specific to N and S proteins of SARS-CoV-2.

An index value (i) corresponds to the division of relative fluorescence values (RFV) by the RFV of the provided standard. The assay is considered negative when i <  1.00 and positive when i  ≥  1.00. Assay sensitivity is 96.6% at ≥16 days after positive rRT-PCR confirmation [10].

### 2.4. Micro-Neutralization (MN) Assay

SARS-CoV-2 was isolated from a positive nasopharyngeal swab during August 2020 and propagated in Vero cells (ATCC^®^ CCL-81™), using complete DMEM supplemented with 1% FBS. Then, 250 µL of the clinical specimen was used to inoculate a 25 cm^2^ cell culture flask. Infectivity was checked with SARS-CoV-2-specific RT-PCR through the reduction of Ct values in the culture supernatant. The virus stock was titrated in 96-well culture plates of Vero cells using 1 log serial dilutions (1 to 11 log) to obtain a 50% tissue culture infective dose (TCID50). Cultures were observed daily using inverted microscope within 3 days for the presence of the cytopathic effect (CPE). The viral titer was expressed in TCID50/mL and calculated using the Spearman and Kärber method.

A day before the neutralization assay, each well was seeded with 20,000 cells, to obtain a 70–80% sub-confluent monolayer after 24 h. The MN assay was performed as previously reported by Grzelak et al. [11]. Briefly, after heat-inactivation, the serum samples were mixed with equal volumes of 100 TCID 50 of SARS-CoV-2 at 2-fold serial dilutions starting from 1:10. The serum-virus mix was incubated for 1 h at 37 °C with 5% CO_2_. After incubation, 100 μL of the mixture at each dilution was passed in duplicate to a 96-well cell plate containing a 70–80% confluent Vero monolayer. A virus back-titration was performed with culture medium replacing the serum to assess the input virus dose. After 3 days of incubation, the plates were inspected under an inverted microscope for CPE. The endpoints of each serum are reported as a serum neutralization titer, which corresponds to the reciprocal of the highest serum dilution that neutralizes the infectious virus using the Spearman and Kärber method as modified by Finney [12]. Samples with a neutralization titer ≥10 were considered positive.

All of the steps manipulating the SARS-CoV-2 and infected cell cultures were carried out at the biosafety level 3 laboratory of the Department of Biosafety PCL3, Laboratory of Research and Medical Analysis, Gendarmerie Royale, Rabat, MA.

### 2.5. Statistics

A comparison of the commercial assay results with the gold standard microneutralization assay was made to assess their performance in detecting NAbs. For sensitivity, calculations were only carried out with microneutralisation positive samples. Negative samples were used to assess specificity and cross-reactivity.

Figures including plotting and receiver operating characteristic (ROC) curves were drawn with Prism (Version 9, GraphPad, San Diego, CA, USA). The convalescent group was considered as such when the serum sample did not pass 14 days after a laboratory-confirmed COVID-19 diagnosis.

*p*-value < 0.05 was considered statistically significant.

## 3. Results

### 3.1. Neutralizing Antibodies to SARS-CoV-2

The serum samples obtained from 177 laboratory workers, including 39 diagnosed as COVID-19 positive during late 2020 and the first quarter of 2021, were assessed using a cell-based virus neutralization test (Figure 2). Neutralizing antibodies (NAb) against SARS-CoV-2 were detected in 149 (84%) of the total number of sera.

The NAb titers were highly variable and ranged between 10 and 640, with a mean ±  SD of 213 ± 187 (median, 160; IQR, 80–320) for convalescent, and a mean  ±  SD of 61  ±  114 (median, 20; IQR 10–40,) for PCR-negative individuals.

Titers of 10 to 40 were categorized as low titers, 80 to 160 as moderate, and ≥320 as high titers. The distribution of the measured neutralization titer is different between the two studied groups after vaccination (PCR-negative individuals and recovered COVID-19 patients) (Figure 3).

The vast majority of the convalescent individuals had moderate-to-high titers of neutralizing antibodies: 11 (28.2%) for 160 and 11 (28.2%) for ≥320 titers (Figure 3 left). Neutralizing antibodies were undetectable in only one convalescent individual(<10).

Vaccinated PCR-negative individuals presented low titers (Figure 3 right). Of 138, the majority (49%) had a titer ranging between 10 and 40, and 14 (10%) between 40 and 80. A total of 27 (19.5%) had no neutralizing antibodyresponse.

The small number (n = 9) (Figure 3 right) of PCR-negative individuals showing a titer ≥320 may have contracted the disease, but their PCR was negative or they were not diagnosed during the infection. They may have felt some of the symptoms of COVID-19 (headache, loss of smell and taste).

### 3.2. Qualitative Serology

In total, 177 samples were examined in parallel comparing SARS-CoV-2 neutralization assay and both rapid LFIA and the automated ELFA mentioned above.

In the rapid LFIA, 149 samples were positive and 28 were negative. The same totals were found in the MN test. In addition, 146 samples were determined to be positive in both cases, and 25 were found to be negative by both assays, resulting in a consensus for 96.61% of the samples. Three samples that were negative in the rapid LFIA were positive in the MN test, and three other samples that were positive in the rapid LFIA were negative in the MN test.

The positive concordance rate of the rapid LFIA was 97.9%, compared with the MN test, while the negative concordance rate was 89.28% (Table 1).

False negative LFIA results were obtained in 1.6% of the patient sera, mainly containing low levels of neutralizing antibodies.

In automated ELFA VIDAS^®^, 168 samples were found to be positive and nine were negative. In total, 148 samples were determined to be positive by both MN and VIDAS^®^ assays, and eight were negative in both assays, which represented a consensus for 88.13% of the samples. The positive concordance (PC) rate of the automated ELFA in comparison with the MN test was 99.32%, while the negative concordance rate (NC) was 28.57%. False positive results were obtained in 20 patient seradue to the presence of other anti-SARS-CoV-2 non-neutralizing antibodies, namely, anti-nucleocapsid and anti-spike proteins (Table 2).

In summary, the qualitative results showed that the sensitivity of the rapid LFIA COVID-PRESTO^®^ (AAZ) was 97.98% (95% confidence interval [CI] 96–100) and the specificity was 89.28% (95% CI 77100). For the VIDAS^®^ SARS-CoV-2 IgG, the sensitivity was 99.33% (95% CI 98100) and the specificity was 28.57% (95% CI 360) when compared to the MN assay as reference method. In both antibody tests, the seropositive specimens revealed a quite good to moderate correlation.

### 3.3. Quantitative Serology

A comparison between the neutralizing antibody titers range in the MN assay and the IgG antibody index levels in the VIDAS^®^ SARS-CoV-2 IgG as well as the score values assigned to the rapid lateral flow was made to explore their correlation. The quantitative results of both commercial methods (177 samples from 177 patients) were plotted against the reciprocal neutralizing titer (Figure 4 and Figure 5).

The two-dimensional distribution diagrams (Figure 4 and Figure 5) show a moderate-to-high correlation with low dispersions of the antibody values within the SN titers. Additionally, median antibody levels increased with increasing neutralizing activity.

Correlation coefficient is employed to describe the strength and direction of the linear association between the neutralizing activity and two SARS-CoV-2 tests. Although both assays showed a positive correlation with neutralizing activity, the strongest (ρ = 0.9341) was found for the rapid lateral flow (AAZ^®^) (Figure 4). The VIDAS SARS-CoV-2 IgG assay (Figure 5) showed a moderate positive correlation (ρ = 0.8995).

In participants with a negative virus neutralization test (<10), the antibody levels vary remarkably in the VIDAS SARS-CoV-2 IgG assay (Figure 5). The absence of neutralization is accompanied by a low antibody level ranging from an index of 1.01 to 3.75 (data not shown). However, we cannot establish a threshold above which neutralization activity is clearly present, due to the overlap of positive and negative values of serum neutralization.

In the case of the rapid lateral flow (AAZ^®^), only 3 of 28 samples with a positive score (score = 1) were not expected to neutralize.

### 3.4. Receiver Operating Characteristics Analysis (ROC)

Finally, receiver operating characteristics (ROC) curves were generated to assess the performance of each serological assay to detect the presence of neutralizing antibodies (NT > 10) (Figure 6).

The areas under the curve (AUC) were 0.97 for the rapid lateral flow (AAZ^®^) and 0.88 for the ELFA VIDAS^®^, which means excellent performances for both immunoassays, but a better performance in estimating the presence of neutralizing antibodies for the first method.

## 4. Discussion

In the first part of the study, the neutralization activity was investigated for 177 serum samples of convalescent and PCR-negative individuals, with both groups immunized with adenovirus-based vaccines. Serostatus data were not available before vaccination, only PCR results were used to distinguish between the convalescents and healthy individuals. The percentage of vaccinated individuals with a positive seroneutralization result was more meaningfully important in the convalescent group than in the PCR-negative group (97.5% vs. 80.5%). We conclude that most of the convalescent individuals have moderate-to-high titers of neutralizing antibodies in comparison to the PCR-negative individuals after 6 to 8 weeks of their second dose. Studies have shown that the NAb response peaks at 3–5 weeks after infection and degrades over 8 monthsfollowing infection. The long-term responses of NAb titers, especially after AstraZeneca vaccination was investigated, demonstrating a possible influence of genderandage. Lim et al. found that NAb titers among the elderly population start to decrease at 8 weeks, and at 16 weeks after the second inoculation [13].

Further studies are needed to monitor post-vaccination immune responses beyond two months and after the third dose to determine the duration of vaccine effectiveness represented by neutralizing activity in particular against emerging variants of SARS-CoV-2. Indeed, some studies have provided assurance of a protective immune response after booster vaccination against SARS-CoV-2 variants [14].

Overall, our results showed that the COVID-19 vaccine improves the level of neutralizing antibodies, and significantly boosts those in individuals naturally infected compared with those with no previous SARS-CoV-2 infection. A previous study showed that the immunity provided by two doses of ChAdOx1 (AstraZeneca/Oxford) is somewhat weaker and declines faster than mRNA vaccines. However, with the combination of infection-induced immunity and vaccine-induced immunity called “hybrid immunity”, neutralizing antibody titers and the extent of SARS-CoV-2 variant recognition are significantly higher in previously infected individuals receiving at least one dose of a COVID-19 vaccine. Moreover, hybrid immunity from vaccination and subsequent infection also results in equally robust immune responses [15]. Indeed, it has been well established that SARS-CoV-2 infection significantly elicits the neutralizing antibody response before or after vaccination in comparison with two doses of vaccine alone [16] and the infection alone delivers temporary protection from COVID-19 [17], confirming the importance of vaccination, regardless of infection history.

Rapid LFIA COVID-PRESTO^®^ (AAZ) and VIDAS^®^ were compared with the microneutralization assay for the qualitative detection of antibodies against SARS-CoV-2. A correlation was found between anti-SARS-CoV-2 IgG response between both methods and neutralizing activities.

In the sensitivity test, both assays demonstrated excellent sensitivity greater than 97%. VIDAS^®^ SARS-CoV-2 IgG showed slightly higher sensitivity than rapid LFIA COVID-PRESTO^®^ (AAZ) when compared to the MN assay. However, the specificity was lower meaning that these assays generated false positive results due to the detection of non-neutralizing antibodies.

In previous studies, the RBD protein provides lower sensitivity and higher specificity than the N protein. A correlation was found between anti-RBD IgG response and neutralizing activities [8].

Considering that N-based serological tests are more sensitive than S protein, while RBD-based serological tests are more specific [18], a better composition of RBD and N protein in serological tests can improve both sensitivity and specificity for forecasting NAb activity.

Moreover, it was recently demonstrated that the VIDAS SARS-CoV-2 test was able to detect virus neutralizing antibodies with perfect concordance (Cohen’s kappa coefficient of 0.9) between the IgG performed in VIDAS and the MN test [19].

Furthermore, the performance of COVID-PRESTO^®^ (AAZ) was evaluated in a clinical study for its specificity and sensitivity compared to a test of reference (RT-PCR) [20]. However, there are no previously published results regarding the correlation with neutralizing activity. Our results showed that rapid LFIA COVID-PRESTO^®^ (AAZ) has a higher specificity (89.28%) to detect NAbs with a reduced number (3/28) of false positive results in comparison with ELFA VIDAS^®^, whose false positives reached 20/28 due to thedetection of non-neutralizing antibodies.

Our results demonstrate a strong positive correlation between the gold standard MN assay and the quantitative results of both immunoassays (LFIA and ELFA) with Spearman’s ρ values ranging from 0.8 to 0.9. The strongest positive correlation was found for the rapid LFIA COVID-PRESTO^®^ (AAZ) assay, with an area under the ROC curve of 0.97, confirming that the rapid test is as an efficient tool to assess neutralizing activity as the MN test.

Previous studies showed that the positivity threshold reported in the instructions for using commercial anti-SARS-CoV-2 serology assays is not a threshold for correlating with neutralization. In order to correlate perfectly with seroneutralization, higher titers of antibodies are needed although this depends on the diversity of the response for each individual [21,22].

Gillot et al. attempted to adapt the cut-offs of some serological assays to improve the capacity of NAbs detection for these assays. However, it was difficult to deal with the loss of specificity or sensitivity to increase other parameters [23].

In particular, when using LFIA, it is necessary to establish a band interpretation system for each laboratory, along with observer training to allow more objective results. We also anticipate that such a serological binding method will play a critical role in SARS-CoV-2 antibody testing and become a convenient routine neutralizing antibody test.

This work establishes the effectiveness of vaccination against a strain that circulated in 2020–2021 and its ability to neutralize the virus. Although the study did not investigate the humoral response to novel variants, other papers have been able to demonstrate vaccine efficacy against SARS-CoV2 mutations, showing that spike-binding and neutralizing activity was maintained and remained unaffected by viral genome variations [24].

Our findings demonstrate a strong positive correlation among SARS-CoV-2 IgG antibody titers in both binding antibody assays (LFIA and ELFA) and neutralizing activity. The strongest positive correlation to neutralizing activity was found for the rapid LFIA. Although this method has only been designed for the qualitative analysis of SARS-CoV-2 antibodies so far, it provides the rapid detection of neutralizing antibodies with high specificity and sensitivity, and thus possesses advantages over conventional microneutralization, which involves the manipulation of the live virus, as well as being a low-cost, equipment-free and on-site test. To the best of our knowledge, no other study has been published that rapid LFIA COVID-PRESTO^®^ (AAZ) assay correlates with the neutralizing antibody response against SARS-CoV-2.

## Figures and Tables

**Figure 1 vaccines-11-00590-f001:**
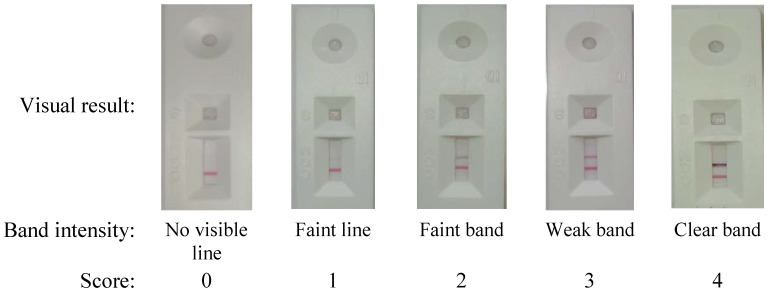
Test line scoring of the qualitative detection of SARS-CoV-2 IgG with COVID-PRESTO^®^(AAZ) based on intensity of lines.

**Figure 2 vaccines-11-00590-f002:**
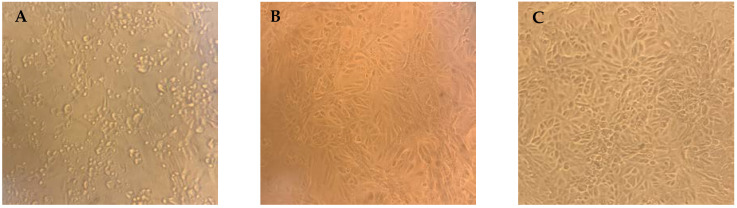
Viral cytopathic effects (CPE) of SARS-CoV-2 on Vero cells and neutralizing antibody activity. (**A**) Non-protective serum showing lysed cells due to viral replication (absence of neutralizing antibodies); (**B**) protective serum at low dilution of 1:10, showing inhibition of CPE by specific neutralizing antibodies; (**C**) highest serum dilution of 1:320 that protected cells from CPE taken as the neutralizing antibody titer.

**Figure 3 vaccines-11-00590-f003:**
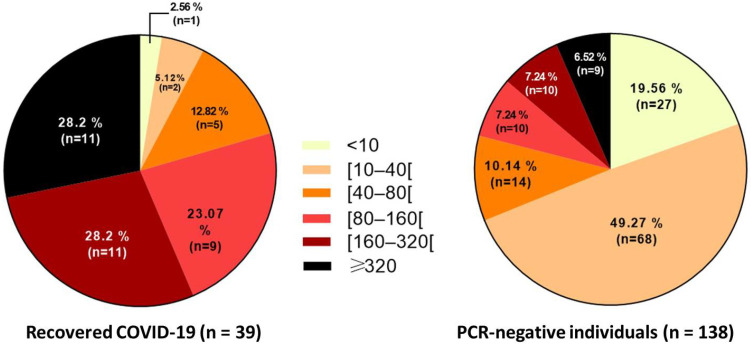
Distribution of neutralizing antibody titers among recovered COVID-19 patients (**left**) and PCR-negative individuals (**right**) after vaccination. The titer values are indicated by a gradient; lighter colors toward darker colors, to indicate the level of neutralizing activities between individuals.

**Figure 4 vaccines-11-00590-f004:**
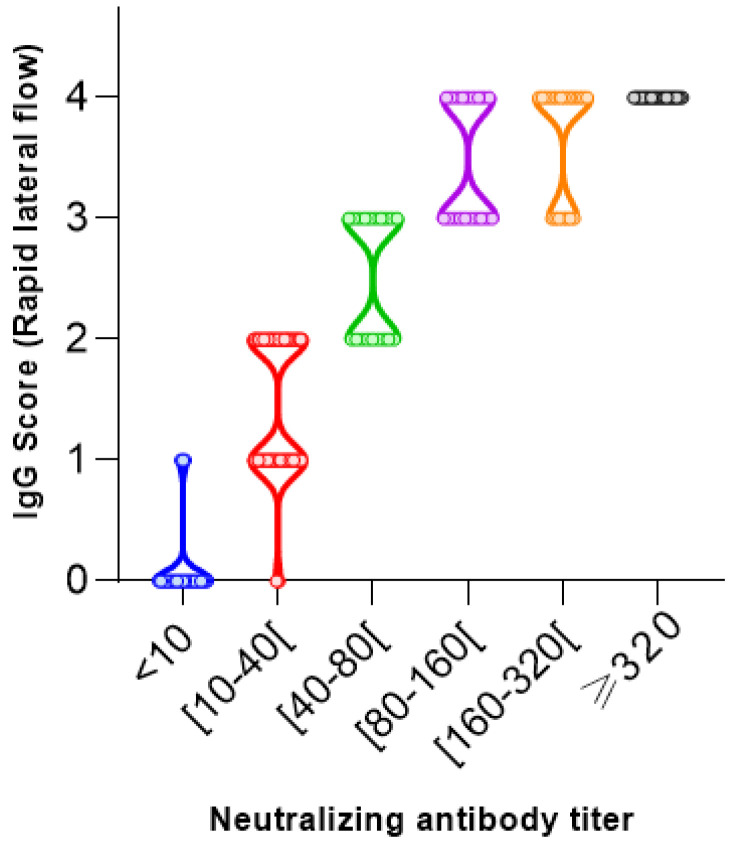
SARS-CoV-2 neutralizing antibody titers correlate with SARS-CoV-2 rapid lateral flow (AAZ^®^) (Spearman correlation coefficients ρ = 0.9341 and *p*-value < 0.0001).

**Figure 5 vaccines-11-00590-f005:**
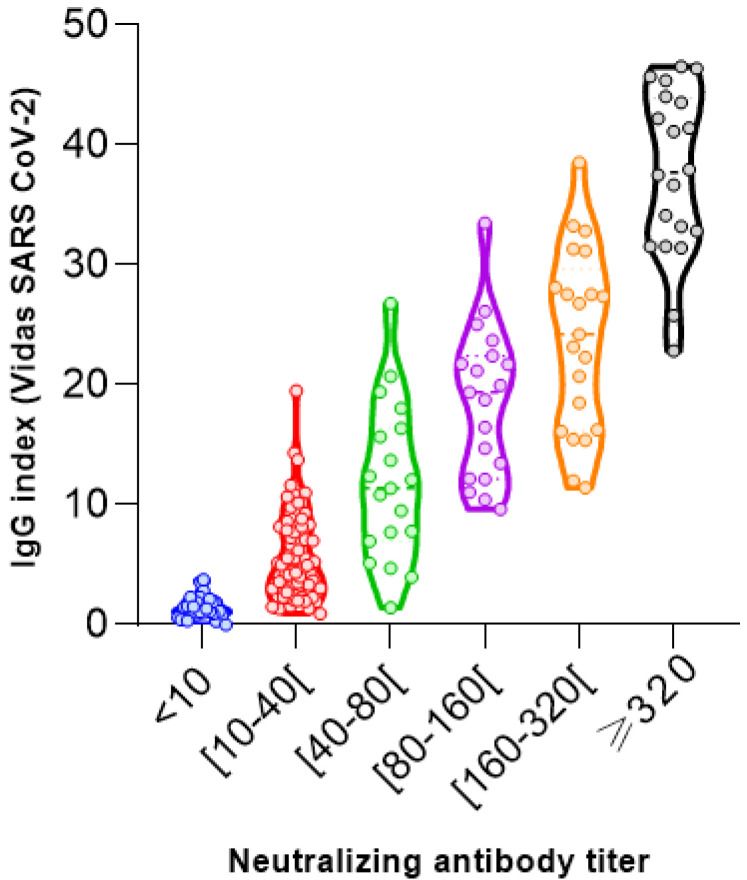
SARS-CoV-2 neutralizing antibody titers correlate with SARS-CoV-2 IgG (VIDAS^®^) (Spearman correlation coefficients ρ = 0.8995 and *p*-value < 0.0001).

**Figure 6 vaccines-11-00590-f006:**
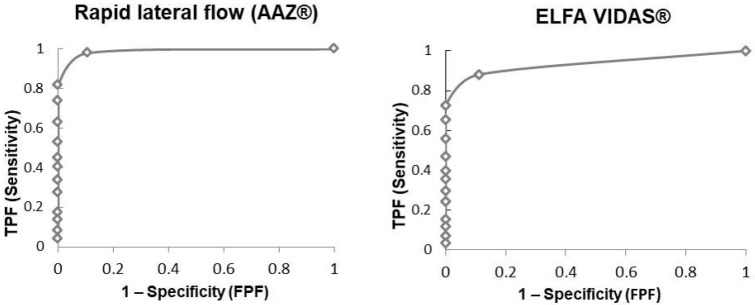
Receiver operating characteristics (ROC) curve analysis for two immunoassays compared to MN assay (titer > 10). TFP: true positive fraction. FPF: false positive fraction.

**Table 1 vaccines-11-00590-t001:** Determination of the concordance of rapid LFIA for the detection of anti-SARS-CoV-2 IgG antibodies to microneutralization assay for the detection of neutralizing antibodies.

Rapid LFIA COVID-PRESTO^®^ (AAZ)	Microneutralization Test		
	Positive	Negative	Total
**Positive**	146	3	149
**Negative**	3	25	28
**Total**	149	28	177

We set a titer of 10 as a limit of detection in the neutralization assay. The absence of a colored band in the test region is a negative result in the rapid LFIA test.

**Table 2 vaccines-11-00590-t002:** Determination of the concordance of VIDAS^®^ SARS-CoV-2 IgG for the detection of anti-SARS-CoV-2 IgG antibodies and microneutralization assay for the detection of neutralizing antibodies.

ELFA VIDAS® SARS-CoV-2 IgG	Microneutralization Test		
	Positive	Negative	Total
**Positive**	148	20	168
**Negative**	1	8	9
**Total**	149	28	177

We set a titer 10 as a limit of detection in neutralization assay, and an index = 1 as a limit of detection of IgG in VIDAS^®^ SARS-CoV-2 test.

## Data Availability

The data that support the findings of this study are available from the corresponding author upon reasonable request.

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
