# Peer review of "SARS-CoV-2-Neutralizing Antibody Response and Correlation of Two Serological Assays with Microneutralization"

_vaccines, 2023, doi:10.3390/vaccines11030590_

Round 1
Reviewer 1 Report
The authors conclude that rapid serological tests used in the city laboratory (ELFA) or home user-friendly (LFIA) are equally effective compared to a gold strandard seroneutralization assay performed in laboratory research. These findings are very interesting since it may allow the quick and inexpensive identification of SARS-CoV-2 level of protection in the population.
However some important details and clarification are required before publication.
Major comments :
What is the viral strain used in this study? Indeed, the authors used sera samples from 39 recovered COVID-19 patients diagnosed positive during late 2020 and the first quarter of 2021. Did the authors use, in the microneutralization tests, the same virus strain that circulated in 2020-2021? What are the viral characteristics of the strain used in LFIA or in VIDAS? Is it the same virus? If these strains are differents, it strongly compromise the interpretation which can preclude the publication of this paper.
Authors state that the same totals samples positive for LFIA ( in line 184) were also positive for the SN assays. Which these samples are the same ? It is important to add a clarification here. A supplemental information (table with details for samples results for SN and serology) should be include in this manuscript.
As authors state in line 202, false positive sera samples were recorded for the ELFA assay (13.5%) and LFIA (10.7%) assay which contradicts the conclusion of the study that claim that rapid serological tests are as efficient as the seroneutalization test. The authors should clarify and discuss about that point in the abstract or discussion section of the manuscript.
Minor comments :
line 67 : "39 recovered COVID-19 patients diagnosed positive" How does these patients were recorded positive ? PCR ? Serology?
line 112 : please harmonize : SARS-CoV-2 or SARS CoV-2
line 113: Vero is not well written
line 179 : indicate the number of sera
line 242: "(=1)". Something is missing here
Table 1 and 2 : Unclear : which results are for the SN assay and which are for the LFIA assay?
Figure 4 and 5 : Why the representations are different (the antibody titers are put on the X-axis in figure 4 and on the Y-axis in figure 5. And why are they not represent as 1/10, 1/20, 1/40 etc...
Why are the antibody titers not represented in a conventional way in Figure 4 (1/10, 1/20, 1/40 etc) as in Figure 5?
Author Response
Response to Reviewer 1 Comments
The authors conclude that rapid serological tests used in the city laboratory (ELFA) or home user-friendly (LFIA) are equally effective compared to a gold strandard seroneutralization assay performed in laboratory research. These findings are very interesting since it may allow the quick and inexpensive identification of SARS-CoV-2 level of protection in the population.
However some important details and clarification are required before publication.
Major comments :
What is the viral strain used in this study? Indeed, the authors used sera samples from 39 recovered COVID-19 patients diagnosed positive during late 2020 and the first quarter of 2021. Did the authors use, in the microneutralization tests, the same virus strain that circulated in 2020-2021? What are the viral characteristics of the strain used in LFIA or in VIDAS? Is it the same virus? If these strains are differents, it strongly compromise the interpretation which can preclude the publication of this paper.
Response : The viral strain that we have used in microneutralization tests for all study have been isolated during August 2020 in the laboratory, propagated in cell culture and titred. The LFIA and VIDAS use the antibody response against the viral infection of subjects that were positive during the same period of late 2020 and the first quarter of 2021.
The date of the isolation of the unique strain used in this study have been added in the text.
Authors state that the same totals samples positive for LFIA ( in line 184) were also positive for the SN assays. Which these samples are the same ? It is important to add a clarification here. A supplemental information (table with details for samples results for SN and serology) should be include in this manuscript.
Response :
No, the samples are not the same, only by coincidence give the same totals. “for another sample combination » was added in the text. The table with details for samples results for SN and serology were included as supplementary material.
As authors state in line 202, false positive sera samples were recorded for the ELFA assay (13.5%) and LFIA (10.7%) assay which contradicts the conclusion of the study that claim that rapid serological tests are as efficient as the seroneutalization test. The authors should clarify and discuss about that point in the abstract or discussion section of the manuscript.
Response : False positive results, means that samples negative in SN are positive in commercial assays (ELFA and LFIA). For rapid serological LFIA the percentage of false positive is lower (3/28) in comparison with ELFA Vidas (20/28). The specificity was also higher in rapid serological LFIA. For the purpose of study, rapid test is efficient for detection of Nabs, rather than ELFA that detect all kinds of antibodies (especially non-neutralizing antibodies).
This was clarified in the text (see discussion)
Minor comments :
line 67 : "39 recovered COVID-19 patients diagnosed positive" How does these patients were recorded positive ? PCR ? Serology?
Response: Positive were recorded by PCR, because there was routine laboratory testing for workers with suspected covid-19 (symptomatic and contacts). These informations were available before starting the study. This was clarified in the text.
line 112 : please harmonize : SARS-CoV-2 or SARS CoV-2
Response: done
line 113: Vero is not well written
Response : done
line 179 : indicate the number of sera
Response : done
line 242: "(=1)". Something is missing here
Response : done
Table 1 and 2 : Unclear : which results are for the SN assay and which are for the LFIA assay?
Response : Results of SN are presented vertically for both figures , and LFIA (table 1) or ELFA Vidas (Table 2) horizontally.
Figure 4 and 5 : Why the representations are different (the antibody titers are put on the X-axis in figure 4 and on the Y-axis in figure 5. And why are they not represent as 1/10, 1/20, 1/40 etc...
Response : Thank you for your comment, the representations have been uniformized, by putting antibody titers in X-axis, and boxes were replaced with violin plot showing all points of values for a better representation of the results.
Why are the antibody titers not represented in a conventional way in Figure 4 (1/10, 1/20, 1/40 etc) as in Figure 5?
Response :
In CPE based microneutralization assay, the neutralizing antibody titer was defined as the reciprocal of the highest serum dilution at which no CPE breakthrough in any of the duplicate testing wells was observed (see material and methods). The titration end point is reported as a reciprocal of the titer (e.g., 1:40 is reported as 40).
This was uniformized in the manuscript.
Reviewer 2 Report
Souiri et al Review Report
Summary
Souiri et al report the results of their work comparing rapid, point of care assays with the more time- and resource-intensive neutralization assay to determine whether the former can provide a useful proxy for the presence of neutralizing antibodies against SARS-CoV-2. They find that there is indeed good correlation between the methods. A simple lateral flow assay analyzed semi-quantitatively in fact provided the best correlation with the microneutralization results. The authors also found evidence SARS-CoV-2 infection may induce a greater quantity of neutralizing antibodies, although clarification on aspects of this are needed to better evaluate this claim.
Main Points
- Given the emphasis on the majority of the subjects being “PCR-negative”, does this mean that the convalescent subjects were PCR-positive at the time the samples were taken? If the authors can provide more detail on some of the timing it may help readers to evaluate their claims regarding infection vs vaccination. This is to say, if the convalescent subjects are all much closer to illness than the 2 months post-vaccination of the others, might this account for the results instead?
- Related to the above point, were the self-reported COVID symptoms mentioned at line 177 exclusive to individuals in the ≥320 titer group? Were all subjects surveyed for such symptoms as a part of the study or did they offer that information to those collecting the samples?
- Could the authors please elaborate on the scoring criteria for the lateral flow assay, particularly the difference between a “line” and “band” distinguishing a score of 1 vs 2. This becomes more relevant @ line 196 (table 1 legend) where it states “absence of a colored band in the test region is a negative result”. Does this mean that scores of both 0 and 1 are negative?
- Some of the results may be more readily interpreted in tabular form. It would also be useful to have vital data on the vaccinees vs convalescents to aid in interpreting the differences in neutralization between them. Depending on whether/how tables are incorporated, fig. 3 might no longer be needed.
Minor Comments
- The naming convention for the neutralization assay and its results changes rather abruptly from MN, which is what is used in the specific section within the methods, and SN (serum neutralization, presumably?). For clarity, it would be best if a single designation were used throughout.
- How strong was the concordance in scores between operators of the lateral flow assay and is there an industry standard that this can be measured against?
- The neutralization bins introduced @ line 163 differ from those used in the figures. Also, it is unclear for the bins used in the figures what side a particular score will fall on (e.g. serum that neutralizes at a 1:80 dilution but not a 1:160 dilution is assigned a score of 80, but does that then count in the 40-80 or 80-160 bin?).
- Please clarify the conclusion @ line 265. Is the intention to say that vaccination is inducing nAbs but that vaccination + infection is inducing more?
- Miscellaneous comments on the introduction
o Could use some additional references, e.g. neutralization as gold standard
o Death toll appears out of date
o Please provide the definition of RBD when it is first used @ line 35
o The statement about serological testing pre- and post-vaccine era would seem to depend heavily on whether a test includes spike, nucleocapsid, or both and testing vaccinees for past infection via nucleocapsid should still be possible
Author Response
Response to Reviewer 2 Comments
Summary
Souiri et al report the results of their work comparing rapid, point of care assays with the more time- and resource-intensive neutralization assay to determine whether the former can provide a useful proxy for the presence of neutralizing antibodies against SARS-CoV-2. They find that there is indeed good correlation between the methods. A simple lateral flow assay analyzed semi-quantitatively in fact provided the best correlation with the microneutralization results. The authors also found evidence SARS-CoV-2 infection may induce a greater quantity of neutralizing antibodies, although clarification on aspects of this are needed to better evaluate this claim.
Main Points
- Given the emphasis on the majority of the subjects being “PCR-negative”, does this mean that the convalescent subjects were PCR-positive at the time the samples were taken? If the authors can provide more detail on some of the timing it may help readers to evaluate their claims regarding infection vs vaccination. This is to say, if the convalescent subjects are all much closer to illness than the 2 months post-vaccination of the others, might this account for the results instead?
Response : This was clarified in the text (see Materials and Methods , 2.1. Specimen collection)
- Related to the above point, were the self-reported COVID symptoms mentioned at line 177 exclusive to individuals in the ≥320 titer group? Were all subjects surveyed for such symptoms as a part of the study or did they offer that information to those collecting the samples?
Response : This information was collected after the results of serological studies, as this group of participants (≥320) are available at the lab, to interpret the results of this few number. For the rest of participants, the symptoms were not surveyed and don’t be a part of the study.
- Could the authors please elaborate on the scoring criteria for the lateral flow assay, particularly the difference between a “line” and “band” distinguishing a score of 1 vs 2. This becomes more relevant @ line 196 (table 1 legend) where it states “absence of a colored band in the test region is a negative result”. Does this mean that scores of both 0 and 1 are negative?
Response : We meant that the “line” is thinner than a “band”. However, for the negative result, we change in the text "No visible band" by No visible line" to make sense.
You can find in the following references a presentation of practically the same scoring approach for a semi-quantitative results to rapid serological tests: 10.1186/s12879-017-2413-x ; 10.1038/s41587-020-0659-0
- Some of the results may be more readily interpreted in tabular form. It would also be useful to have vital data on the vaccinees vs convalescents to aid in interpreting the differences in neutralization between them. Depending on whether/how tables are incorporated, fig. 3 might no longer be needed.
Response : We added a legend in Figure 3, precising the “recoverd Covid-19” and “pcr-negative”. Categories of neutralization levels are indicated with a gradient color (from yellow, for negative, to stronger color red and black for high titers). This have been specified in the legend of figure 3. A table will all data have been incorporated as supplementary material.
Minor Comments
- The naming convention for the neutralization assay and its results changes rather abruptly from MN, which is what is used in the specific section within the methods, and SN (serum neutralization, presumably?). For clarity, it would be best if a single designation were used throughout.
Response : We made corrections in the text by keeping MN for the name of the assay, and SN to indicate serum neutralization (titer, level…)
- How strong was the concordance in scores between operators of the lateral flow assay and is there an industry standard that this can be measured against?
Response : The concordance between both operators was 100%, as the reading is based on captured photos of scores.
In the text we state that this scoring was performed for research purposes to capture semi-quantitative data about the rapid test, and its laboratory-dependant. This was previously described in many papers (see reference [7]):
- Whitman JD, Hiatt J, Mowery CT, et al. Test performance evaluation of SARS-CoV-2 serological assays. medRxiv; 2020. DOI: 10.1101/2020.04.25.20074856. (see reference [7])
- Routsias, J.G., Mavrouli, M., Tsoplou, P. et al. Diagnostic performance of rapid antigen tests (RATs) for SARS-CoV-2 and their efficacy in monitoring the infectiousness of COVID-19 patients. Sci Rep 11, 22863 (2021). https://doi.org/10.1038/s41598-021-02197-z
- The neutralization bins introduced @ line 163 differ from those used in the figures. Also, it is unclear for the bins used in the figures what side a particular score will fall on (e.g. serum that neutralizes at a 1:80 dilution but not a 1:160 dilution is assigned a score of 80, but does that then count in the 40-80 or 80-160 bin?).
Response: Bins have been reviewed and corrected in the text to be in accordance with bins in figures.
80 is inside the bin between the brackets [80-160[, where the 80 is included, and 160 is excluded
- Please clarify the conclusion @ line 265. Is the intention to say that vaccination is inducing nAbs but that vaccination + infection is inducing more?
Response : The sentence have been reviewed.
- Miscellaneous comments on the introduction
o Could use some additional references, e.g. neutralization as gold standard
o Death toll appears out of date
o Please provide the definition of RBD when it is first used @ line 35
Response :
- The reference number 6 was added (Liu et al., 2022) for SN as a gold standard.
- Death toll have been updated to 7 February 2023 (OMS)
- receptor-binding domain have been added for RBD
o The statement about serological testing pre- and post-vaccine era would seem to depend heavily on whether a test includes spike, nucleocapsid, or both and testing vaccinees for past infection via nucleocapsid should still be possible
- This statement has been clarified
Reviewer 3 Report
The manuscript under review is an interesting work on the correlation between microneutralization and two serological assays (LFIA and ELFA). The manuscript is interesting, but I have some comments for the Authors.
- The paper needs a whole-manuscript English revision.
- In the introduction, line 30, when referring to COVID-19, I think “This was a fatal infection” is not correct, because COVID-19 could still be a fatal infection. I suggest re-writing this sentence.
- In the introduction, line 60, please clarify what “MNT” stand for, because it is the first time it is mentioned within the text.
- I think the Authors should deepen a little bit the introduction when explaining the use of rapid tests (lines 49-53). I suggest precising the mechanisms of these tests (what they detect, etc etc).
- In the introduction, I suggest explaining why it is important to investigate the laboratory staff seroprevalence (and not the patients seroprevalence, for example).
- In materials and Methods, 2.1 specimen collection, lines 69-70, “at least 2 months after last dose of the vaccination”, which is the maximum range of time after the last dose? At least 2 months-maximum 8 months/1 year/etc etc after the vaccination? In the discussion, the authors wrote “6 to 8 weeks”, is this the time period after the vaccination the samples have been collected?
- In materials and Methods, 2.1 specimen collection, lines 73-74, “Participants gave oral informed consent about the study results that would not influence any clinical decisions about their specific case.”, so the consent was only orally taken?
- In materials and Methods, 2.1 specimen collection, I suggest precising that all the included people were negative for SARS-CoV-2 at the time of the sample collection.
- I suggest implementing the paper with a table summarizing the characteristic of the study group (for example, how many males and females, how many people had had covid19, how many of them were vaccinated, etc etc).
- I think the references list needs to be implemented, for example with the following papers: 10.1016/j.ebiom.2021.103626; 10.3390/v14081644; 10.1038/s41598-022-16097-3
Author Response
Response to Reviewer 3 Comments
The manuscript under review is an interesting work on the correlation between microneutralization and two serological assays (LFIA and ELFA). The manuscript is interesting, but I have some comments for the Authors.
- The paper needs a whole-manuscript English revision.
Response : we suggest to transfer the manuscript to an English editing service.
- In the introduction, line 30, when referring to COVID-19, I think “This was a fatal infection” is not correct, because COVID-19 could still be a fatal infection. I suggest re-writing this sentence.
Response : done
- In the introduction, line 60, please clarify what “MNT” stand for, because it is the first time it is mentioned within the text.
Response : done
- I think the Authors should deepen a little bit the introduction when explaining the use of rapid tests (lines 49-53). I suggest precising the mechanisms of these tests (what they detect, etc etc).
Response : The use of rapid tests have been clarified in the text.
- In the introduction, I suggest explaining why it is important to investigate the laboratory staff seroprevalence (and not the patients seroprevalence, for example).
Response : Laboratory workers data were recorded over time and were easily available to lunch this study (PCR results, history of infection, serological testing) and represent a part of population (including patients) that were in the first line who received immunization against COVID-19 with two doses in same conditions.
- In materials and Methods, 2.1 specimen collection, lines 69-70, “at least 2 months after last dose of the vaccination”, which is the maximum range of time after the last dose? At least 2 months-maximum 8 months/1 year/etc etc after the vaccination? In the discussion, the authors wrote “6 to 8 weeks”, is this the time period after the vaccination the samples have been collected?
Response : Thank you for your comment, the specimen collection was after 6 to 8 weeks of vaccination. This was clarified in the text.
- In materials and Methods, 2.1 specimen collection, lines 73-74, “Participants gave oral informed consent about the study results that would not influence any clinical decisions about their specific case.”, so the consent was only orally taken?
Response : Yes, the participant gave their oral informed consent and were invited to blood collection and they are aware of the importance of such investigation about the immune response as they are in a medical profession. Under the general rules regarding data protection, the contact details of the subjects were kept confidential, and after collecting the samples, the names were deleted and replaced by patient codes.
- In materials and Methods, 2.1 specimen collection, I suggest precising that all the included people were negative for SARS-CoV-2 at the time of the sample collection.
Response : Indeed, this was precised in the text
- I suggest implementing the paper with a table summarizing the characteristic of the study group (for example, how many males and females, how many people had had covid19, how many of them were vaccinated, etc etc).
Response : We included this information as a supplementary material, however the gender is not a part of result interpretations.
- I think the references list needs to be implemented, for example with the following papers: 10.1016/j.ebiom.2021.103626; 10.3390/v14081644; 10.1038/s41598-022-16097-3
Response :
Thank you for your suggestions, the first reference has been cited in the discussion.
Unfortunately, the second reference which discusses the safety of certain vaccines and their impact on health, in particular cardiovascular events, is far from the purpose of our study.
The third reference have been mentioned, even the durability of immunity beyond two months couldn’t be monitored for this study.
Round 2
Reviewer 1 Report
Major comment n°1 :
The authors response regarding the strain caracteristics’s is not sufficient. Did the authors performed a whole genome sequencing of that isolate ? At August 2020, several variants of concerns were already circulating (original strain from Wuhan, Alpha, Beta). This should be absolutely investigated since these variants carrying further mutations in their spike proteins that raise concerns regarding their evasion to neutralizing antibodies induced by vaccination or primary infection. This is admit by several studies ( ex : PMID : 34237773; 36434068)
Major comment n°2 :
This is confusing. Indeed, if the positive samples for MN are not the same in LFIA, what is the relevance of such results and interpreation ? This is unclear and contradictory with the supplemental excel data in which we can clearly see that 146 out of 177 same samples are positive for MN and LFIA. The author explanation is confuse. Their sentence in the manuscript leads to confusion and need to be removed.
Author Response
Major comment n°1 :
The authors response regarding the strain caracteristics’s is not sufficient. Did the authors performed a whole genome sequencing of that isolate ? At August 2020, several variants of concerns were already circulating (original strain from Wuhan, Alpha, Beta). This should be absolutely investigated since these variants carrying further mutations in their spike proteins that raise concerns regarding their evasion to neutralizing antibodies induced by vaccination or primary infection. This is admit by several studies ( ex : PMID : 34237773; 36434068)
Response : Indeed, the strain has been sequenced and analyzed by our team. It’s a wild type, with no presence of any specific mutations of VOC.
The sequencing have shown the presence of a spike mutation D614G, that predominated globally since April and May 2020 witch increase infectivity but not alter S-protein binding to ACE2 or neutralization sensitivity, as demonstrated by (Zhang et al., 2020) (PMID 33243994).
The sequence has not been published yet and it may be included in an upcoming paper from the group.
Major comment n°2 :
This is confusing. Indeed, if the positive samples for MN are not the same in LFIA, what is the relevance of such results and interpreation ? This is unclear and contradictory with the supplemental excel data in which we can clearly see that 146 out of 177 same samples are positive for MN and LFIA. The author explanation is confuse. Their sentence in the manuscript leads to confusion and need to be removed.
The sentence only describes the results in table 1, concerning Rapid LFIA and neutralization assay. In rapid LFIA, 149 samples were positive and 28 were negative. In MN, 149 samples were positive and 28 were negative. “Same totals” was used in the text to not repeat values.
Of course, the same samples (n=146) are positive in MN and LFIA. But the total of 149 that are positive in MN are not the same positives in LFIA, it’s evident because the concordance was not 100%. The sentence added in the previous revision is confusing, so we removed it.
Reviewer 3 Report
The manuscript has improved after the revisions, I think it can be published after a whole manuscript English revision
Author Response
Thank you for your comment, however, the english still need to be improved, i will send the manuscript to the MDPI english editing.